# Alcohol withdrawal syndrome in ICU patients: Clinical features, management, and outcome predictors

**Aliénor Vigouroux**☯, **Charlotte Garret**☯, **Jean-Baptiste Lascarrou**☯, **Maëlle Martin**☯, **Arnaud-Félix Miailhe**☯, **Jérémie Lemarié**☯, **Julien Dupeyrat**☯, **Olivier Zambon**☯, **Amélie Seguin**☯, **Jean Reignier**[iD]☯, **Emmanuel Canet**[iD]*☯

Service de Médecine Intensive Réanimation, Centre Hospitalier Universitaire de Nantes, Université de Nantes, Nantes, France

☯ These authors contributed equally to this work.
* emmanuel.canet@chu-nantes.fr

**Data Availability Statement:** All relevant data are within the paper and its Supporting Information files.

## Abstract

### Background

Alcohol withdrawal syndrome (AWS) is a common condition in hospitalized patients, yet its epidemiology in the ICU remains poorly characterized.

### Methods

Retrospective cohort of patients admitted to the Nantes University Hospital ICU between January 1, 2017, and December 31, 2019, and coded for AWS using ICD-10 criteria. The objective of the study was to identify factors associated with complicated hospital stay defined as ICU length of stay ≥7 days or hospital mortality.

### Results

Among 5,641 patients admitted to the ICU during the study period, 246 (4.4%) were coded as having AWS. Among them, 42 had exclusion criteria and 204 were included in the study. The three main reasons for ICU admission were sepsis (29.9%), altered consciousness (29.4%), and seizures (24%). At ICU admission, median Cushman's score was 6 [4–9] and median SOFA score was 3 [2–6]. Delirium tremens occurred in half the patients, seizures in one fifth and pneumonia in one third. Overall, 48% of patients developed complicated hospital stay, of whom 92.8% stayed in the ICU for ≥7 days, 36.7% received MV for ≥7 days, and 16.3% died during hospital stay. By multivariable analysis, two factors were associated with complicated hospital stay: a higher number of organ dysfunctions at ICU admission was associated with a higher risk of complicated hospital stay (OR, 1.18; 95CI, 1.05–1.32, *P* = 0.005), whereas ICU admission for seizures was associated with a lower risk of complicated hospital stay (OR, 0.14; 95%CI, 0.026–0.80; *P* = 0.026).

**Funding:** The author(s) received no specific funding for this work.

**Competing interests:** Emmanuel Canet received fees for lectures and conference talks and had travel and accommodation expenses to attend scientific meetings covered by Gilead, Baxter, and Sanofi-Genzyme. The other authors declare that they have no competing interests. This does not alter our adherence to PLOS ONE policies on sharing data and materials.

**Abbreviations:** AWS, alcohol withdrawal syndrome; CI, confidence Interval; CIWA, Clinical Institute of Withdrawal Assessment for alcohol scale; DSM-5, Diagnostic and Statistical Manual of Mental Disorders, fifth edition; ICD-10, International Classification of Diseases, 10th revision; ICU, intensive care unit; IQR, interquartile range; MV, mechanical ventilation; OR, odds ratio; SAPS, Simplified Acute Physiology Score; SOFA, Sepsis-related Organ Failure Assessment.

## Conclusions

AWS in ICU patients chiefly affects young adults and is often associated with additional factors such as sepsis, trauma, or surgery. Half the patients experienced an extended ICU stay or death during the hospital stay. The likelihood of developing complicated hospital stay relied on the reason for ICU admission and the number of organ dysfunctions at ICU admission.

## Introduction

Alcohol is the most commonly used psychoactive substance in adults and a major cause of hospitalization, morbidity, and mortality worldwide. In 2016, 32.5% of the world's population were current drinkers and 2.8 million deaths were attributed to alcohol use [1]. In France, an estimated 5 million people have alcohol-use disorder [2], and alcohol is a leading risk factor for premature death and disability [3]. One of the adverse consequences of chronic alcohol use is alcohol withdrawal syndrome (AWS). AWS may lead to acute neurotoxicity due to an extensive release of glutamate neurotransmitters and a massive opening of post-synaptic calcium channels which induces neuronal apoptosis [4]. Among heavy alcohol users, approximately 50% experience some degree of withdrawal symptoms when their consumption is reduced or stopped [5–8] and about 10% have withdrawal seizures [5–7]. Moreover, AWS can progress to delirium tremens, a state characterized by severe confusion and hallucinations associated with severe autonomic hyperactivity [5–7]. The most severe forms of AWS may require ICU admission, and a study conducted in Finland found that 20% of ICU admissions were related to alcohol use [9]. Several studies have attempted to identify risk factors for developing AWS and delirium tremens [6,10–18], while others focused on the therapeutic strategy [17–24]. However, the epidemiology of AWS in ICU patients is poorly known, its optimal management remains chiefly empirical, and its outcome is largely unstudied. A better understanding of the epidemiology, treatment, and outcome in ICU patients with AWS may help guide clinical practice and research.

We therefore conducted an epidemiological study in a French university-affiliated ICU, by using the International Classification of Diseases 10th Revision (ICD-10) coding system to identify patients with AWS. We aimed to test the hypothesis that AWS in ICU patients could result in extended ICU stay or death, and to identify factors associated with such outcomes.

## Methods

This retrospective study was approved by the ethics committee of the French Intensive Care Society (CE SRLF 21–10) on February 01, 2021 with a waiver for informed consent. The study is reported in compliance with the STROBE recommendations [25].

### Study design, setting, and population

We identified consecutive adults (≥18 years of age) admitted to the ICU of the Nantes University Hospital between January 1, 2017, and December 31, 2019, and registered in the electronic hospital database with any of the codes for AWS in the ICD-10 (F10.3, F10.30, F10.31, F10.4, F10.40, F10.41, F10.03, F10.05, F10.06). For patients who had multiple admissions during the study period, only the first admission was considered. In our institution, coding is done at the time of ICU discharge by the physician in charge, using the patient's formal discharge

summary. Each medical file was reviewed by AV and EC to confirm the diagnosis of AWS based on the Diagnostic and Statistical Manual of Mental Disorders, Fifth Edition (DSM-5) [26]. All four major criteria had to be present in the electronic medical record of each patient to diagnose AWS. For major criterion B, 2 or more of the 8 symptoms had to be present. Data were extracted from the doctors and nurses notes (S1 Fig). Delirium tremens was defined as a patient with AWS who developed a state of confusion, acute agitation and hallucinations recorded in the medical file during the ICU stay. Exclusion criteria were absence of signs and/ or symptoms of AWS recorded in the medical file, withdrawal syndrome of unclear origin (absence of chronic alcoholism or withdrawal of a substance other than alcohol), and preventive treatment for AWS without subsequent AWS.

## Data collection

Data were extracted from the electronic medical records of the ICU (CERNER Millenium®, Nantes, France). We obtained data for baseline patient characteristics, including demographics, comorbidities, chronic medications, and habits of alcohol consumption. Habits of alcohol consumption and alcohol history are part of the standard intake procedure in our hospital. Data were obtained from patients' interview. When the patient's clinical condition made the interview impossible, data were obtained either from the next of kin or from the patient at the time of discharge. The onset of AWS was the date when AWS was first recorded in the medical file. For each patient, AWS recovery was assessed by reading the daily notes of nurses and doctors from the EMR. The date of resolution was either the date of resolution recorded in the medical file or the last date of recording of AWS with no further signs or symptoms of AWS recorded for at least 48 hours. If neither of these two conditions was met, the episode of AWS was classified as persistent. When patients had underlying dementia or other neurocognitive disorders, a worsening of the clinical state during hospitalization had to be mentioned in the patients' EMR to classify a patient with persistent confusion. AWS severity was assessed using Cushman's score [27]. The following complications of AWS during the ICU stay were recorded: delirium tremens, seizures, status epilepticus, rhabdomyolysis, acute kidney injury, and pneumonia. The drugs administered intravenously or orally during the AWS episode were extracted from the electronic prescription database of the hospital. The life-sustaining therapies used during the ICU stay (high-flow oxygen, non-invasive ventilation, mechanical ventilation (MV), vasopressors, and/or renal replacement therapy) were extracted from the electronic medical records. Vital status and destination at hospital discharge (home, psychiatry ward, rehabilitation center, or discharge against medical advice) were also recorded.

## Objectives

The primary objective of the study was to identify factors associated with complicated hospital stay. Complicated hospital stay was defined as ICU length of stay ≥7 days or death before hospital discharge. In the absence of these criteria, patients were considered to have uncomplicated hospital stay.

The secondary objectives were to describe the clinical features, treatments, and outcomes of ICU patients with AWS.

## Statistical analysis

Continuous variables are described as median and interquartile range [IQR] and compared using Wilcoxon's test. Categorical variables are described as counts (percent) and compared using the exact Fisher's test. The occurrence of complicated hospital stay (versus uncomplicated hospital stay) was analyzed as a binary variable. Logistic regression analyses were

performed to identify variables associated with complicated hospital stay, with estimated odds ratios (ORs) and their 95% confidence intervals (95%CIs). For the multivariable model, we preselected candidate variables which plausibly fit with complicated hospital stay based on knowledge from the literature (SOFA, comorbidities, and mortality) and our assumptions (chronic use of BZD or antipsychotics, history of AWS, reason for ICU admission, and extended stay in the ICU). We carefully checked to avoid collinearity between variables and we applied the rule of selecting a maximum of 1 variable per 8 events (total of 12 variables in our study). All tests were two-sided, and P values lower than 5% were considered to indicate significant associations. Statistical tests were conducted using the R statistics program, version 3.5.0 (R Foundation for Statistical Computing, Vienna, Austria; www.R-project.org/).

## Results

### Study population

Among 5,641 patients admitted to the ICU during the study period, 246 (4.4%) were coded as having AWS. A detailed analysis of the medical files showed that 42 patients had exclusion criteria (Fig 1). The remaining 204 patients were included in the study (Fig 1). Table 1 reports their main features. Median daily alcohol intake was 129 (72–216) grams (missing data, n = 51). Patients were admitted from the emergency department (67.2%), wards (19%), or pre-hospital emergency medical service (14%). Computed tomography (CT) of the brain was performed in 77 (37.8%) patients, of whom 22 (28.6%) had the following abnormal findings: subarachnoid bleeding, n = 6; subdural hematoma, n = 6; stroke, n = 4; intracranial hematoma, n = 2; extra-dural hematoma, n = 2; chronic hydrocephalus, n = 1; and cortical atrophy compatible with Wernicke encephalopathy, n = 1.

### Clinical features of AWS and ICU management

AWS was typically diagnosed 1 [1–2] day after ICU admission and lasted 5 [3–8] days. Table 2 provides details about the treatments used and complications observed. All patients were treated with a combination of vitamins B1 and B6 and intravenous hydration during the first

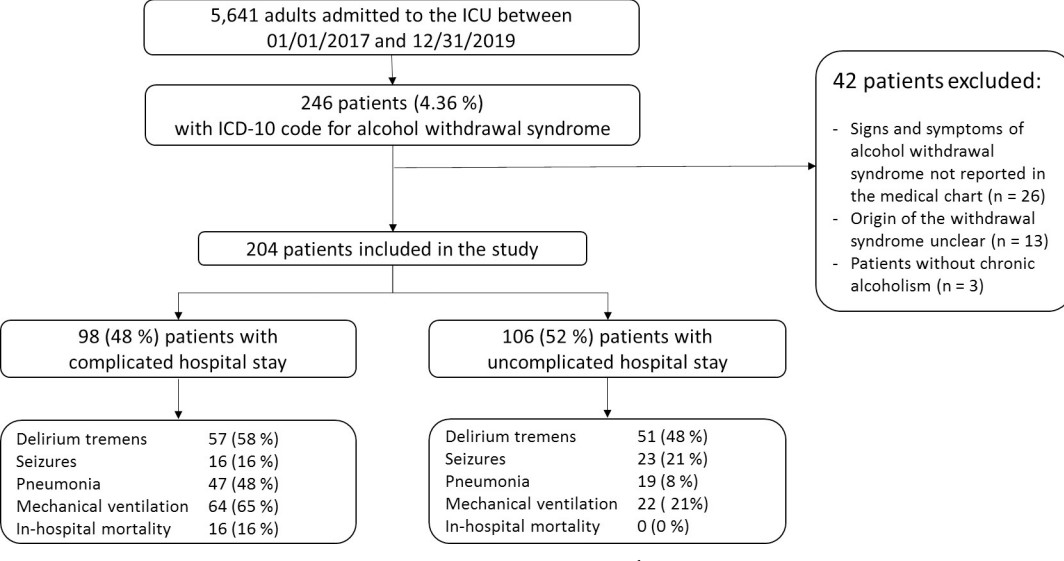

**Fig 1. Study flowchart.** ICD-10: International Classification of Diseases 10<sup>th</sup> Revision.

**Table 1.  Baseline characteristics of the 204 study participants.**

| Variable | All patients (n = 204) | Complicated hospital stay (n = 98) | Uncomplicated hospital stay (n = 106) | P value |
|---|---|---|---|---|
| **Demographics** | | | | |
| Age, median [IQR], years | 53 [46–60] | 54.5 [48–61] | 50 [44–58] | 0.099 |
| Male sex, n (%) | 172 (84.3) | 83 (84.7) | 89 (84.0) | 0.88 |
| Charlson's index, median [IQR] | 1 [0–3] | 2 [0.25–4] | 1 [0–3] | 0.019 |
| **Alcohol withdrawal history** | | | | |
| History of AWS, n (%) | 42 (20.6) | 23 (23.5) | 19 (18.0) | 0.387 |
| History of DT, n (%) | 10 (4.9) | 7 (7.1) | 3 (2.8) | 0.200 |
| History of withdrawal seizures, n (%) | 25 (12.3) | 15 (15.3) | 10 (9.4) | 0.201 |
| **Psychiatric history** | | | | |
| Substance use disorder other than alcohol, n (%) | 30 (14.7) | 12 (12.2) | 18 (17.0) | 0.429 |
| Any psychiatric disorder, n (%)[a] | 71 (34.8) | 33 (33.7) | 38 (35.9) | 0.598 |
| Mood disorders, n (%) | 7 (3.4) | 4 (4.1) | 3 (2.8) | 0.712 |
| Anxiety disorders, n (%) | 64 (31.4) | 29 (29.6) | 35 (33) | 0.651 |
| **Chronic medications** | | | | |
| Benzodiazepines, n (%) | 71 (34.8) | 27 (27.6) | 44 (41.5) | 0.036 |
| Antipsychotic drugs, n (%) | 26 (12.8) | 6 (6.1) | 21 (18.9) | 0.0063 |
| **Time from hospital admission to ICU admission, median [IQR], days** | 0 [0–1] | 0 [0–1] | 0 [0–1] | 0.168 |
| **ICU admission from the ED, n (%)** | 137 (67.2) | 63 (64.3) | 74 (69.8) | 0.401 |
| Cushman's score at ED admission, median [IQR] | 7 [4–9] | 7 [4–9] | 7 [4–9] | 0.827 |
| **Reason for ICU admission, n (%)** | | | | 0.00073 |
| Sepsis | 61 (29.9) | 39 (39.8) | 22 (20.8) | |
| Altered consciousness | 60 (29.4) | 22 (22.5) | 38 (35.9) | |
| Seizures | 24 (11.7) | 4 (4.1) | 20 (18.9) | |
| Trauma | 20 (9.8) | 9 (9.2) | 11 (10.4) | |
| Surgery | 12 (5.9) | 6 (6.1) | 6 (5.7) | |
| AKI | 12 (5.9) | 7 (7.2) | 5 (4.7) | |
| Other[b] | 15 (7.4) | 11 (11.2) | 4 (3.8) | |
| **Clinical variables and measures at ICU admission** | | | | |
| HR, median [IQR], bpm | 104 [88–120] | 109 [98–123] | 99 [85–117] | 0.038 |
| SBP, median [IQR], mmHg | 125 [106–147] | 121 [100–140] | 126 [112–150] | 0.038 |
| Glasgow Coma Scale score, median [IQR] | 14 [12–15] | 14 [11–15] | 14 [13–15] | 0.84 |
| RR, median [IQR] | 22 [18–26] | 22.5 [19.3–27] | 20 [17–25] | 0.015 |
| Cushman score | 6 [4–9] | 6 [4–9] | 7 [4–9] | 0.196 |
| SOFA | 3 [2–6] | 5 [3–8] | 3 [1–5] | <0.0001 |
| SAPS II | 24 [16–34] | 27 [17–37] | 20 [13–31] | 0.0067 |

AKI: Acute kidney injury; AWS: Alcohol withdrawal syndrome; BPM: Beats per minute; DT: Delirium tremens; ED: Emergency department; HR: Heart rate; ICU: Intensive care unit; IQR: Interquartile range; RR: Respiratory rate; SAPS II: Simplified Acute Physiology Score, version II; SBP: Systolic blood pressure; SOFA: Sequential Organ Failure Assessment.

[a]Any psychiatric disorder, n (%): Including 6 patients with underlying dementia.

[b]Other: Cardiac or respiratory arrest; upper gastrointestinal hemorrhage; acute pancreatitis; ketoacidosis; mesenteric ischemia.

24 hours. Benzodiazepines were given to 99% of patients. Diazepam and oxazepam were often given intermittently. Continuous midazolam or propofol were prescribed to 27.9% and 12.3% of the patients, respectively. Among patients with pneumonia, the most commonly recovered micro-organisms were *Streptococcus pneumoniae*, *methicillin-sensitive Staphylococcus aureus*,

**Table 2. Clinical features of AWS and pharmacological management.**

| Variable | All patients (n = 204) | Complicated hospital stay (n = 98) | Uncomplicated hospital stay (n = 106) | P value |
|---|---|---|---|---|
| **Clinical features of AWS** | | | | |
| Time from ICU admission to AWS onset, days, median [IQR] | 1 [1–2] | 1 [1–3] | 1 [1–2] | 0.6083 |
| Worst Cushman score during the ICU stay, median [IQR] | 11 [8–14] | 12 [9–15] | 11 [8–13] | 0.099 |
| IV fluids during the first 24 h, L median [IQR] | 2 [1.5–3] | 2 [1.5–3] | 2 [1.5–3] | 0.484 |
| B1 and B6 vitamin therapy, n (%) | 204 (100) | 98 (100) | 106 (100) | 1.00 |
| **Drugs administered at the time of AWS** | | | | |
| **Benzodiazepines** | | | | |
| Diazepam (IV/PO), n (%) | 170 (83.3) | 82 (83.7) | 88 (83) | 0.9003 |
| Oxazepam (PO), n (%) | 102 (50) | 52 (53.1) | 50 (47.2) | 0.4005 |
| Midazolam (continuous IV), n (%) | 57 (27.9) | 39 (39.8) | 18 (17) | 0.0028 |
| Length of treatment with BZD, days, median [IQR] | 4 [3–7] | 7 [4–9] | 3 [2–4] | <0.0001 |
| **Antipsychotics** | | | | |
| Haloperidol (IV) | 26 (12.3) | 13 (13.3) | 13 (12.3) | 0.83 |
| Cyamemazine (PO) | 37 (18.1) | 25 (25.5) | 12 (11.3) | 0.0085 |
| Loxapine (IM) | 6 (2.9) | 5 (5.1) | 1 (0.9) | 0.1056 |
| **Other drug** | | | | |
| Propofol (continuous IV) | 25 (12.3) | 22 (22.5) | 3 (2.8) | <0.0001 |
| **AWS-related diagnoses, n (%)** | | | | |
| Delirium tremens | 108 (52.9) | 57 (58.2) | 51 (48.1) | 0.15 |
| Seizures | 39 (19.1) | 16 (16.3) | 23 (21.7) | 0.329 |
| Status epilepticus | 18 (8.8) | 5 (5.1) | 13 (12.3) | 0.071 |
| Pneumonia | 66 (32.4) | 47 (48) | 19 (17.9) | <0.0001 |
| AKI | 60 (29.4) | 31 (31.6) | 29 (27.4) | 0.503 |
| Rhabdomyolysis | 12 (5.9) | 7 (7.1) | 5 (4.7) | 0.461 |
| **Duration of AWS, days, median [IQR]** | 5 [3–8] | 8 [6.25–13] | 4 [3–5] | <0.0001 |
| **Life-sustaining therapies** | | | | |
| High flow oxygen, n (%) | 9 (4.4) | 5 (5.1) | 4 (3.8) | 0.644 |
| Non-invasive ventilation, n (%) | 16 (7.8) | 13 (13.3) | 3 (2.8) | 0.0056 |
| MV, n (%) | 86 (42.2) | 64 (65.3) | 22 (20.8) | <0.0001 |
| MV duration, days, median [IQR] | 5.5 [2–10] | 7.50 [4–12.3] | 2 [1.25–2] | <0.0001 |
| Vasopressors, n (%) | 33 (16.2) | 27 (27.6) | 6 (5.7) | <0.0001 |
| Renal replacement therapy, n (%) | 2 (1) | 2 (2) | 0 (0) | 0.2295 |

AKI: Acute kidney injury; AWS: Alcohol withdrawal syndrome; BZD: Benzodiazepine; ICU: Intensive care unit; IM: Intramuscular; IQR: Interquartile range; IV: Intravenous; MV: Mechanical ventilation; PO: Per os.

and *Haemophilus influenzae*. In the patients who required MV, the time interval between the diagnosis of AWS and endotracheal intubation ranged from -1 to +2 days after the diagnosis of AWS. The most common reasons for intubation were coma (75.6%) and acute respiratory failure (24.4%).

## Outcomes

During the study period, the occurrence of complicated hospital stay in ICU patients was 48% in patients with AWS and 31% in patients without AWS (p<0.001, S1 Table). Among AWS patients who developed complicated hospital stay, 92.8% stayed in the ICU for ≥7 days, 36.7%

**Table 3. Outcomes.**

| Variable | All patients (n = 204) | Complicated hospital stay (n = 98) | Uncomplicated hospital stay (n = 106) | *P* value |
|---|---|---|---|---|
| **AWS outcome** | | | | |
| Persistent confusion at ICU discharge, n (%) | 51 (25) | 34 (34.7) | 17 (16.0) | 0.0003 |
| Persistent agitation at ICU discharge, n (%) | 12 (5.9) | 8 (8.2) | 4 (3.8) | 0.120 |
| **Length of stay** | | | | |
| ICU, days, median [IQR] | 6 [4–10.3] | 11 [8–15.8] | 4 [3–5] | <0.0001 |
| Hospital, days, median [IQR] | 13 [7–26.3] | 23 [13.5–34] | 9 [5–14] | 0.0085 |
| **Vital status** | | | | |
| ICU mortality, n (%) | 11 (5.4) | 11 (11.2) | 0 (0) | 0.00039 |
| Hospital mortality, n (%) | 16 (7.8) | 16 (16.3) | 0 (0) | <0.0001 |
| **Destination at hospital discharge[a]** | | | | |
| Home, n (%) | 118 (57.8) | 51 (52.0) | 67 (63.2) | 0.959 |
| Follow-up care and rehabilitation unit, n (%) | 29 (14.2) | 20 (20.4) | 9 (8.5) | 0.0022 |
| Psychiatric ward, n (%) | 16 (7.8) | 3 (3.1) | 13 (12.7) | 0.037 |
| Left against medical advice, n (%) | 7 (3.4) | 1 (1.0) | 6 (5.7) | 0.113 |
| Addictology unit, n (%) | 3 (1.47) | 0 (0) | 3 (2.8) | 0.126 |
| **Alive on day 28, n (%)** | 189 (92.7) | 83.9 (85.6) | 106 (100) | 0.00027 |

AWS: Alcohol withdrawal syndrome; IQR: Interquartile range; ICU: Intensive care unit; LOS: Length of stay.

[a]: Missing data n = 31 (15.2%).

required MV for ≥7 days, and 16.3% died during the hospital stay (Table 3). The duration of AWS in patients with complicated hospital stay was twice that in patients with uncomplicated hospital stay (8 [6–13] versus 4 [3–5] days). Patients with complicated hospital stay were twice as likely to have persistent confusion at ICU discharge and had more than twice the hospital length of stay, compared to patients with uncomplicated hospital stay. Finally, the destination at hospital discharge differed between the two groups (Table 3).

## Factors associated with complicated hospital stay

By univariate analysis, comorbidities, organ dysfunctions at ICU admission, tachycardia, low blood pressure, high respiratory rate, and sepsis were associated with an increased risk of developing complicated hospital stay. In contrast, chronic use of benzodiazepines or neuroleptic drugs was more common in patients with uncomplicated hospital stay. By multivariable analysis, only two factors were independently associated with developing complicated hospital stay: a higher number of organ dysfunctions at ICU admission was associated with a higher risk of complicated hospital stay, while ICU admission for seizures was associated with a lower risk of complicated hospital stay (Table 4). In another multivariable model which included all patients admitted to the ICU during the study period, AWS was by itself a factor associated with a higher risk of complicated hospital stay (S2 Table).

## Discussion

### Key findings

We used the ICD-10 coding system and DSM-5 criteria to identify ICU patients who developed AWS. We found that patients with AWS accounted for approximately 4% of all ICU admissions and that half of them developed complicated hospital stay despite having low severity scores at ICU admission. Furthermore, patients with complicated hospital stay had more co-morbidities, were more likely to be admitted for sepsis, displayed higher SOFA scores at

**Table 4. Logistic regression analyses for factors associated with complicated hospital stay.**

| Factors | Univariate analysis | | Multivariable analysis[a] | |
|---|---|---|---|---|
| | OR (95%CI) | P value | OR (95%CI) | P value |
| **Demographics** | | | | |
| Age (per year) | 1.02 (0.99–1.05) | 0.10 | | |
| Male sex | 0.95 (0.44–2.02) | 0.87 | | |
| Charlson's index | 1.15 (1.02–1.31) | 0.02 | 1.09 (0.95–1.23) | 0.24 |
| **Alcohol withdrawal history** | | | | |
| History of AWS | 1.39 (0.71–2.73) | 0.33 | 1.95 (0.87–4.57) | 0.11 |
| **Chronic medications** | | | | |
| Benzodiazepines | 0.54 (0.30–0.97) | 0.04 | 0.58 (0.29–1.15) | 0.12 |
| Antipsychotic drugs | 0.28 (0.11–0.73) | 0.009 | 0.35 (0.11–0.98) | 0.056 |
| **Cushman's score at ED admission** | 1.01 (0.92–1.10) | 0.826 | | |
| **Reason for ICU admission** | | | | |
| Altered consciousness | 1 | | 1 | |
| Sepsis | 3.06 (1.45–6.45) | 0.031 | 2.02 (0.91–4.58) | 0.088 |
| Seizures | 0.35 (0.10–1.15) | 0.081 | 0.22 (0.05–0.071) | 0.019 |
| Trauma | 1.41 (0.50–3.97) | 0.51 | 1.47 (0.48–4.32) | 0.48 |
| Surgery | 1.73 (0.49–6.06) | 0.39 | 0.93 (0.23–3.74) | 0.92 |
| AKI | 2.42 (0.68–8.61) | 0.17 | 1.50 (0.37–6.31) | 0.57 |
| Other[b] | 4.75 (1.34–16.86) | 0.015 | 2.77 (0.74–12.0) | 0.14 |
| **Clinical variables and measures at ICU admission** | | | | |
| SOFA at ICU admission | 1.23 (1.12–1.36) | 0.00004 | 1.18 (1.06–1.33) | 0.005 |
| HR | 1.01 (1.00–1.02) | 0.041 | | |
| SBP | 0.99 (0.98–0.99) | 0.041 | | |
| RR | 1.05 (1.01–1.09) | 0.018 | | |

AKI: Acute kidney injury; AWS: Alcohol withdrawal syndrome; CI: Confidence interval; ED: Emergency department; HR: Heart rate; ICU: Intensive care unit; OR: Odds ratio; RR: Respiratory rate; SBP: Systolic blood pressure; SOFA: Sepsis-related Organ Failure Assessment.

[a]Preselected candidate variables included in the multivariable model were: Charlson's index, history of AWS, chronic use of benzodiazepines, chronic use of antipsychotic drugs, reason for ICU admission, and SOFA at ICU admission.

[b]Other: Cardiac or respiratory arrest; upper gastrointestinal hemorrhage; acute pancreatitis; ketoacidosis; mesenteric ischemia.

ICU admission, and were more likely to require follow-up care or rehabilitation at hospital discharge compared to patients with uncomplicated AWS. Finally, the likelihood of developing complicated hospital stay was lower in patients with seizures and higher in patients with a higher number of organ failures at ICU admission. Neither Cushman's score nor the occurrence of delirium tremens was associated with the risk of complicated hospital stay.

## Comparison with previous studies

The epidemiology of AWS in ICU patients is difficult to ascertain. The available studies were conducted in specific populations (emergency departments, addiction units, psychiatry wards, trauma centers, or medical wards). They found highly variable incidences ranging from 0.3% to 52% [8,24,28,29]. In a recent review, the incidence of AWS in the ICU patients ranged from <1% in unselected patients to 60% in highly selected alcohol-dependent patients [24]. Studies differed in the tools they used to diagnose and assess AWS, making comparisons difficult. We used both ICD-10 codes and DSM-5 criteria to identify AWS in an unselected population admitted to a university-affiliated ICU. According to these criteria, approximately 4% of ICU patients had AWS.

Male sex, older age, heavier drinking, past history of AWS or withdrawal seizures, greater severity of AWS at hospital admission, concurrent substance use disorder, and mental health conditions have been reported to be associated with a higher risk of developing severe AWS or delirium tremens [6,8,24]. We found that 85% of patients were males and heavy drinkers and that one-third had underlying psychiatric disorders, whereas only a fifth had a history of AWS. Approximately half the patients had sepsis, trauma, or surgery identified as a precipitating factor for AWS, in keeping with previous studies [24,30].

The optimal management of AWS has yet to be determined. Benzodiazepines are considered the cornerstone of therapy despite the lack of a high level of evidence [31], with symptom-triggered bolus administration being the recommended modality [24,32]. In addition, short-acting antipsychotics or alpha2-agonists are often required to treat agitation and autonomic hyperactivity [19,21,33]. In a study conducted in three US hospitals, as many as 16 different medications and 74 combinations of medications were used to treat AWS [21]. In our study, intermittent administration of diazepam or oxazepam was the first-line treatment in nearly all the patients, a continuous infusion of midazolam or propofol was added in nearly 30% of patients, and antipsychotics were used in one fourth of patients. However, our data cannot allow conclusions about the effectiveness of specific treatments on patient outcomes, highlighting the need for further trials. Interestingly, a recent study reported that the implementation of a hospital-wide protocol for the management of AWS resulted in significant improvements in quality of care, decreased the need for ICU admission and the rate of intubation, reduced hospital length of stay, and was cost-savings [34].

The assessment of AWS severity is important to identify patients at high risk for adverse outcomes and to adjust the pharmacological interventions accordingly. However, the definition and assessment of severe AWS has varied across studies [35]. Most studies used scales to grade clinical symptoms, such as the revised Clinical Institute Withdrawal Assessment for Alcohol scale (CIWA-Ar) [5,35,36]. However, this scale was not developed and has not been validated for ICU patients. Moreover, the CIWA-Ar excludes mechanically ventilated patients and has limited accuracy for predicting severe AWS and its complications [35]. We used Cushman's scale [27] to evaluate the severity of AWS. Unlike the CIWA-Ar, Cushman's scale can be used in uncooperative ICU patients. In our study, Cushman's scores obtained at several time points were not associated with the need for prolonged MV, an extended ICU stay, or mortality.

The severity of AWS is often defined as the occurrence of seizures and delirium tremens. However, whether seizures or delirium tremens are associated with clinically relevant adverse outcomes such as a prolonged ICU stay or mortality is unclear [8,24]. In our study, delirium tremens was diagnosed in half of our patients but was not associated with prolonged MV, an extended ICU stay, or mortality. In contrast, we found ICU admission for seizures to be associated with uncomplicated hospital stay. Seizures can result in substantial complications, including status epilepticus or aspiration pneumonia [8]. However, withdrawal seizures can be efficiently treated with benzodiazepines with a potential rapid improvement of clinical status compared to other reasons for ICU admission such as sepsis, acute pancreatitis or acute kidney injury which are more complex to treat. AWS can result in significant morbidity, including aspiration pneumonia, acute kidney injury, and arrhythmia [37]. In historical studies, mortality rates of up to 15% were observed [38,39]. However, in a recent study conducted among trauma patients, AWS-associated mortality was 7% [28]. Our experience was similar, with an overall ICU mortality of 5.4%.

Data on the long-term outcomes of ICU patients with AWS are limited. In a Spanish study, 72% of ICU patients with delirium tremens were readmitted multiple times to the emergency department within the next 2 years [40]. Although AWS is by definition an acute syndrome, a

quarter of our patients had persistent confusion at ICU discharge, more than 40% were unable to return home at hospital discharge, and 7% died within 28 days of ICU admission.

## Study implications

The findings from our study imply that AWS in ICU patients is often triggered by a precipitating factor, with sepsis being the most commonly reported. Therefore, patients with AWS should be routinely screened for sepsis to identify those who require early investigations and treatment. Moreover, our findings imply that, although Cushman's score may help clinicians to titrate the treatment of AWS, it is unable to identify patients at risk for an extended ICU stay, or hospital mortality. In contrast, early detection of organ dysfunctions identifies a population at high risk for adverse outcomes. Thus, the prompt identification, regular re-assessment, and early treatment of organ dysfunctions may improve patient outcomes. Finally, the high frequency of persistent confusion at ICU discharge, high proportion of patients who could not be discharged home, and significant mortality within 28 days after ICU admission support the view that closer surveillance of this vulnerable population may be justified.

## Strengths and limitations

This study has a number of strengths. First, we used both the ICD-10 coding system and a detailed review of each medical file for DSM-5 criteria to identify patients with AWS. This minimized potential bias related to the retrospective selection of the study patients. Second, we evaluated and identified risk factors for clinically relevant endpoints (ICU stay ≥7 days and in-hospital mortality). Thus, we provide new data for identifying patients at risk for poor outcomes. Third, we obtained detailed information on the outcome after ICU discharge, an area rarely explored in previous ICU studies.

Our study also has several limitations. First, the retrospective design implies information bias with a possibility of missing data. For example, we had no information on other potential causes of health disparities, such as income, health care coverage or country of birth, which may have influenced patients' outcomes. Second, the study was conducted in a single institution, where the case mix may have significantly influenced our findings. Nonetheless, we conducted this study in an unselected ICU population in a large university-affiliated center, and our results should therefore apply to similar settings in high-income countries. Third, although ICD-10 discharge coding combined with DSM-5 criteria has strong reliability for diagnosing AWS, sensitivity may be limited [41]. We therefore may have underestimated the true incidence of AWS and studied a particular cohort of patients with more easily diagnosed and, perhaps, more severe and prolonged AWS. However, there is no consensus on the best method for identifying AWS. Finally, the lack of a standardized protocol for managing AWS resulted in substantial variability in the drugs used, their dosages, and their combinations. This prevented us from evaluating how treatments may have affected patient outcomes. However, there is no agreement on the optimal pharmacological treatment of AWS. Current recommendations rely mostly on expert opinion with a low level of evidence [31,42–48].

## Conclusion

In conclusion, ICU patients in this sample drawn from a single hospital in France were predominantly male (84%) with a median age of 53 (IQR 46–60) and were commonly admitted with additional diagnoses including sepsis, trauma, or following elective or urgent surgery. Despite having low severity scores at ICU admission, half the patients experienced an extended ICU stay, or death during hospital stay. The likelihood of developing complicated hospital stay was lower in patients with seizures and higher in those with multiple organ dysfunctions at

ICU admission. A previous history of AWS, Cushman's score, and delirium tremens were not associated with outcomes. These findings suggest that early identification of organ dysfunctions and prompt recognition and treatment of sepsis may improve patient outcomes. Additional trials are needed to determine the optimal therapeutic strategy for decreasing the morbidity and mortality of AWS. Finally, the high frequency of persistent confusion at ICU discharge underlines the need for studies focusing on the long-term outcomes of AWS.

## Supporting information

**S1 Fig. Diagnostic criteria for alcohol withdrawal syndrome according to the Diagnostic and Statistical Manual of Mental Disorders (Fifth Edition) (DSM-5).**
(DOCX)

**S1 Table. Comparison of ICU patients with and without AWS during the study period.**
(DOCX)

**S2 Table. Logistic regression analyses for factors associated with complicated hospital stay among the 5,641 patients admitted to the ICU during the study period.**
(DOCX)

**S1 Dataset.**
(CSV)

## Acknowledgments

We thank Antoinette Wolfe for assistance in preparing and reviewing the manuscript.

## Author Contributions

**Conceptualization:** Aliénor Vigouroux, Charlotte Garret, Jean-Baptiste Lascarrou, Maëlle Martin, Arnaud-Félix Miailhe, Jérémie Lemarié, Julien Dupeyrat, Olivier Zambon, Amélie Seguin, Jean Reignier, Emmanuel Canet.

**Data curation:** Aliénor Vigouroux, Maëlle Martin, Arnaud-Félix Miailhe, Jérémie Lemarié, Julien Dupeyrat, Olivier Zambon, Amélie Seguin.

**Investigation:** Aliénor Vigouroux.

**Methodology:** Aliénor Vigouroux, Emmanuel Canet.

**Supervision:** Jean Reignier, Emmanuel Canet.

**Validation:** Charlotte Garret.

**Writing – original draft:** Aliénor Vigouroux.

**Writing – review & editing:** Charlotte Garret, Jean-Baptiste Lascarrou, Jean Reignier, Emmanuel Canet.

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
