## [Decision Letter · Decision Letter 0]

2 Nov 2021

PONE-D-21-30656Alcohol Withdrawal Syndrome in ICU Patients: Clinical Features, Management, and Outcome PredictorsPLOS ONE

Dear Dr. Canet,

Thank you for submitting your manuscript to PLOS ONE. After careful consideration, we feel that it has merit but does not fully meet PLOS ONE’s publication criteria as it currently stands. Therefore, we invite you to submit a revised version of the manuscript that addresses the points raised during the review process. Please submit your revised manuscript by Dec 17 2021 11:59PM. If you will need more time than this to complete your revisions, please reply to this message or contact the journal office at plosone@plos.org. Please include the following items when submitting your revised manuscript:A rebuttal letter that responds to each point raised by the academic editor and reviewer(s). You should upload this letter as a separate file labeled 'Response to Reviewers'.A marked-up copy of your manuscript that highlights changes made to the original version. You should upload this as a separate file labeled 'Revised Manuscript with Track Changes'.An unmarked version of your revised paper without tracked changes. You should upload this as a separate file labeled 'Manuscript'.

We look forward to receiving your revised manuscript.

Kind regards,

Aleksandar R. Zivkovic

Academic Editor

PLOS ONE

Journal Requirements:

2. Please provide additional details regarding participant consent. In the Methods section, please ensure that you have specified (1) whether consent was informed and (2) what type you obtained (for instance, written or verbal). If your study included minors, state whether you obtained consent from parents or guardians. If the need for consent was waived by the ethics committee, please include this information.

[EC received fees for lectures and conference talks and had travel and accommodation expenses to attend scientific meetings covered by Gilead, Baxter, and Sanofi-Genzyme. 

The authors declare that they have no competing interests.] 

Reviewers' comments:

Reviewer #1: 1) Table 1: Why is patient demographics is limited to age + gender? Health disparities may influence outcomes. Income, access to care, native vs. foreign-born, primary language may factors influencing time of presentation

2) Table 1: complicated v. uncomplicated appear to be 2 different patient populations based on chronic medication use (BZD, antipsychotics). Cessation of BZD can result in seizures. Antipsychotics may lower seizure threshold. Both may result in apparent alcohol withdrawal seizures.

3) Table 2: What was time from hospital admission to ICU admission? Medications received prior to ICU? Treatment prior to ICU arrival may influence ICU outcomes (see #2). For instance, oversedation in ED or hospital wards may lead to aspiration, PNA, sepsis, etc. See Melkonian et al (2019)

4) Table 3: how was persistent confusion at ICU discharge assessed? CAMS-ICU? Clinical impression? How much was delirium vs. dementia? What was baseline confusional status?

5) Discussion: based on multivariable analysis, authors argue that sicker patients (MOD) have complicated stays. It would be interesting to learn how this compares to patients without AWS.

6) Abstract: based on multivariable analysis, authors also argue that seizures are protective. I find this statement illogical since seizures are, by definition, harmful.

7) Discussion: the presence of DTs or alcohol withdrawal seizures may suggest opportunities to standardize treatment of AWS in the ED and general wards, which may hopefully reduce incidence of ICU admission.

Reviewer #2: PONE-D-21-30656 — Alcohol Withdrawal Syndrome in ICU Patients: Clinical Features, Management, and Outcome Predictors

The authors perform a retrospective cohort study of ICU patients with AWS, using manual chart review for data extraction. The main objective was to describe a wide array of factors associated with ICU stay ≥ 7 days and/or in-hospital mortality – a combined outcome the authors labeled “complicated hospital stay.” This is mainly a descriptive study, including descriptions of many patient-level factors stratified by the primary outcome (i.e., complicated hospital stay): demographical characteristics, baseline diagnoses, chronic medications, reason for ICU admission, clinical features associated with acute illness (e.g., SOFA scores), AWS therapies, AWS-related clinical scores (i.e., Cushman) and diagnoses (e.g., seizures), duration of AWS, life-sustaining ICU therapies (e.g., mechanical ventilation), persistent confusion at ICU discharge, length of stay, mortality, and disposition upon hospital discharge. Logistic regression analyses were also performed to further evaluate the association between certain patient factors and the primary composite outcome, but it is not entirely clear how/why the much smaller list of independent variables were selected and included in the model.

The authors should be commended for investigating a grossly understudied yet common ICU condition, offering insights regarding the basic epidemiology of AWS in the ICU and “real world” treatment approaches and hospital course. Although the study is interesting, it lacks focus and does not seem to be driven by a central hypothesis or research question. As a result, the reader gets lost. Many conclusions are stated throughout the discussion that cannot be drawn from this study. Given these broad issues, I am offering general feedback with some specific examples below. Overall, I think the study design and resulting manuscript needs significant restructuring.

Examples of issues that need to be addressed:

Lack of transparency re. methods – the data for this study was mainly obtained via manual chart review. This could be a strength if rigorously approached but was incompletely described. As it stands, the methods section leaves many questions unaddressed. For example, were “habits of alcohol consumption” obtained via patient interview? Is the alcohol history part of the standard hospital intake procedure? What was the frequency of missing data (many ICU patients are too sick to provide history)? Did authors AV and EC manually extract ALL data for the study or just confirm the diagnosis of AWS? How were DSM-5 criteria for AWS applied to information recorded in electronic medical records that was not necessarily designed for assessing these criteria? For example, how was “increased hand tremor” (included in the DSM-5 criteria) assessed via the electronic medical records?

Statistical analysis section is confusing – “Quantitative” and “qualitative” variables are referred to in the first sentence of this section—do the authors mean continuous and categorical variables? It seems a statistical approach (“significant” variables in univariate analyses) versus a hypothesis driven approach was used to design the multivariable logistic regression model. Presumably the authors are using this multivariable model to address confounding, but the underlying hypothesis regarding how these variables relate is unclear. An excellent reference for thinking about study design and presentation of results is: Lederer et al. Annals Am Thorac Soc 2019;16(1):22-28.

“Overreaching” conclusions – In the final paragraph (and similarly stated in the abstract), “AWS in ICU patients chiefly affects young patients with few comorbidities and is often triggered by a precipitating factor such as sepsis, trauma, or surgery.” A more accurate statement from my perspective might be: “ICU patients in this sample drawn from a single hospital in France were predominantly male (84%) with a median age of 53 (IQR 46-60) and were commonly admitted with additional comorbidities including sepsis, trauma, or following (elective?) surgery.” We do not know that AWS was “triggered” by comorbid conditions like sepsis. AWS is more likely triggered by heavy alcohol use that is ALSO possibly associated with these other conditions (based on data from other studies).

Implicit comparisons to a larger ICU sample, not included in the study – The authors seem to make comparisons to a broader ICU cohort. For example, “Despite having low severity scores at ICU admission”… or “the high frequency of persistent confusion” – these statements imply comparisons but the comparison group (implicitly, ICU patients at large) is not defined for the reader. The authors’ tendency to make such comparisons illustrates perhaps the fundamental design flaw of the study. Descriptions of ICU patients with AWS are provided, but without context. The reader is left wondering, how does this compare to “average” ICU patients at the study hospital? The association identified between organ dysfunction and the combined outcome of ICU stay ≥ 7 days and/or in-hospital mortality is not surprising in ICU patients. Whether or not AWS modifies this relationship would be the interesting question; for example, testing the hypothesis that the known association between organ dysfunction and ICU length of stay and/or in-hospital mortality is more pronounced in patients with AWS compared to patients without AWS.

---

## [Author Response · Author response to Decision Letter 0]

24 Nov 2021

PONE-D-21-30656

Alcohol Withdrawal Syndrome in ICU Patients: Clinical Features, Management, and Outcome Predictors

PLOS ONE

Dear Dr. Canet,

Thank you for submitting your manuscript to PLOS ONE. After careful consideration, we feel that it has merit but does not fully meet PLOS ONE’s publication criteria as it currently stands. Therefore, we invite you to submit a revised version of the manuscript that addresses the points raised during the review process.

We look forward to receiving your revised manuscript.

Kind regards,

Aleksandar R. Zivkovic

Academic Editor

PLOS ONE

Journal Requirements:

Author’s reply: The manuscript has been revised and now fulfilled all PLOS ONE’s style requirements. 

2. Please provide additional details regarding participant consent. In the Methods section, please ensure that you have specified (1) whether consent was informed and (2) what type you obtained (for instance, written or verbal). If your study included minors, state whether you obtained consent from parents or guardians. If the need for consent was waived by the ethics committee, please include this information.

Author’s reply: The requested information is now detailed in the revised version of the manuscript. 

[EC received fees for lectures and conference talks and had travel and accommodation expenses to attend scientific meetings covered by Gilead, Baxter, and Sanofi-Genzyme. 

The authors declare that they have no competing interests.] 

Author’s reply: The requested information has been added in the revised version of the manuscript.

Author’s reply: We made changes to our Data Availability statement. This information has been added to the cover letter.

Reviewers' comments:

Reviewer #1: 

Author’s reply: We would like to thank the reviewer for making constructive comments that have helped us to clarify and improve our manuscript. 

1) Table 1: Why is patient demographics is limited to age + gender? Health disparities may influence outcomes. Income, access to care, native vs. foreign-born, primary language may factors influencing time of presentation

Author’s reply: The reviewer raises an important point. We agree that other social and demographic factors may influence patients’ outcome. We report age, gender and the burden of comorbidities measured by the Charlson’s index, but the risk of uncaptured confounding factors is significant. Unfortunately, in our institution, the number of demographic factors routinely recorded in the electronic medical health records is limited. Data related to income, place of birth and primary language were not available. Access to care in public hospitals is unrestricted in France, even for patients not covered by the statutory French health insurance (which account for a very limited number of patients). Although we agree that such patients may experience issues related to follow-up after hospital discharge and access to chronic treatments (key determinants of long-term outcome), our purpose was to focus on the ICU setting and the short-term (day-28 after ICU). 

This limitation has been clearly acknowledged in the revised version of the manuscript (discussion section, limitations) as follow:

“Our study also has several limitations. First, the retrospective design implies information bias with a possibility of missing important data. For example, we had no information on other potential causes of health disparities, such as income, health care coverage or country of birth, which may have influenced patients’ outcomes.”

2) Table 1: complicated v. uncomplicated appear to be 2 different patient populations based on chronic medication use (BZD, antipsychotics). Cessation of BZD can result in seizures. Antipsychotics may lower seizure threshold. Both may result in apparent alcohol withdrawal seizures.

Author’s reply: We agree that these 2 populations share similarities (age, AWS and psychiatric history) but also have significant differences (chronic medications, reason for ICU admission). Although we agree that BZD withdrawal syndrome and the use of antipsychotics increase the risk of seizure, in our study patients who had a complicated hospital stay were four times less likely to be admitted to the ICU for seizures than patients who had an uncomplicated hospital stay (4.1% versus 18.9%, table 1). Therefore, we hypothesize that patients who had such chronic medications did not experience significant BZD deficiency or antipsychotics overdose. Thus, we believe that AWS was not “overdiagnosed” in patients with a complicated hospital stay. Moreover, as AWS is a difficult clinical diagnosis without gold standard, we carefully double checked the plausibility of AWS diagnosis using the combination of ICD10 criteria and DSM-5 criteria with a thorough analysis of each electronic medical record by 2 investigators (AV and EC). Patients who had isolated seizures without the other DSM-5 criteria were not classified as having AWS. Using such methodology, 42 patients were excluded from the analysis because the diagnosis of AWS was unclear (Figure 1, flowchart).

This important point has been clarified in the revised version of the manuscript (methods section) as follow:

“Each medical file was reviewed by AV and EC to confirm the diagnosis of AWS based on the Diagnostic and Statistical Manual of Mental Disorders, Fifth Edition (DSM-5). All four major criteria had to be present in the electronic medical record of each patient to diagnose AWS. For major criterion B, 2 or more of the 8 symptoms had to be present. Data were extracted from the doctors and nurses notes (Supplementary appendix, Figure 1)”. 

3) Table 2: What was time from hospital admission to ICU admission? Medications received prior to ICU? Treatment prior to ICU arrival may influence ICU outcomes (see #2). For instance, oversedation in ED or hospital wards may lead to aspiration, PNA, sepsis, etc. See Melkonian et al (2019)

Author’s reply: Time from hospital admission to ICU admission was 0[0-1] day without significant difference between the complicated group (0[0-1] day) and the uncomplicated group (0[0-1] day) (p=0.168). Of note, more than two-thirds of the patients were admitted to the ICU directly from the ED. In the subgroup of patients who were admitted to the ICU from the medical or surgical wards, median time from hospital to ICU admission was 0[0-2.5] day, without significant difference between the complicated and uncomplicated groups (1[0-3] vs 0[0-1.25], p=0.143). Unfortunately, medications received prior to ICU were not available. Taken all together, our results suggest that time from hospital to ICU admission had no influence on patients’ outcome in our study. However, our results are undoubtedly influenced by the single-centre design of our study, and thus by the policy of ICU referral and admission in our hospital. Even if we do not have predefined criteria for MET/RRT activation, our policy strongly encourages early assessment of all unstable patients by the intensivist and early ICU admission when ED or ward staff are worried about a patient. Therefore, all AWS patients not responding to a first line of treatment (intermittent administration of BZD) are admitted to the ICU and treatment escalation is not administered outside the ICU. 

As suggested by the reviewer, we have added the information in the revised version of the manuscript (Table 1) and the suggested reference in the discussion section. 

Page 15 lines 297-300 

Variable

All patients

 (n = 204)

Complicated hospital stay

(n = 98) 

Uncomplicated hospital stay

 (n = 106) 

P value

Demographics 

Age, median [IQR], years 53 [46-60] 54.5 [48-61] 50 [44-58] 0.099

Male sex, n (%) 172 (84.3) 83 (84.7) 89 (84.0) 0.88

Charlson’s index, median [IQR] 1 [0-3] 2 [0.25-4] 1 [0-3] 0.019

Alcohol withdrawal history 

History of AWS, n (%) 42 (20.6) 23 (23.5) 19 (18.0) 0.387

History of DT, n (%) 10 (4.9) 7 (7.1) 3 (2.8) 0.200

History of withdrawal seizures, n (%) 25 (12.3) 15 (15.3) 10 (9.4) 0.201

Psychiatric history 

Substance use disorder other than alcohol, n (%) 30 (14.7) 12 (12.2) 18 (17.0) 0.429

Any psychiatric disorder, n (%) 71 (34.8) 33 (33.7) 38 (35.9) 0.598

Mood disorders, n (%) 7 (3.4) 4 (4.1) 3 (2.8) 0.712

Anxiety disorders, n (%) 64 (31.4) 29 (29.6) 35 (33) 0.651

Chronic medications 

Benzodiazepines, n (%) 71 (34.8) 27 (27.6) 44 (41.5) 0.036

Antipsychotic drugs, n (%) 26 (12.8) 6 (6.1) 21 (18.9) 0.0063

Time from hospital admission to ICU admission, median [IQR], days 0 [0-1] 0 [0-1] 0 [0-1] 0.168

ICU admission from the ED, n (%) 137 (67.2) 63 (64.3) 74 (69.8) 0.401

Cushman’s score at ED admission, median [IQR] 7 [4-9] 7 [4-9] 7 [4-9] 0.827

Reason for ICU admission, n (%) 0.00073

Sepsis 61 (29.9) 39 (39.8) 22 (20.8) 

Altered consciousness 60 (29.4) 22 (22.5) 38 (35.9) 

Seizures 24 (11.7) 4 (4.1) 20 (18.9) 

Trauma 20 (9.8) 9 (9.2) 11 (10.4) 

Surgery 12 (5.9) 6 (6.1) 6 (5.7) 

AKI 12 (5.9) 7 (7.2) 5 (4.7) 

Other* 15 (7.4) 11 (11.2) 4 (3.8) 

Clinical variables and measures at ICU admission 

HR, median [IQR], bpm 104 [88-120] 109 [98-123] 99 [85-117] 0.038

SBP, median [IQR], mmHg 125 [106-147] 121 [100-140] 126 [112-150] 0.038

Glasgow Coma Scale score, median [IQR] 14 [12-15] 14 [11-15] 14 [13-15]

 0.84

RR, median [IQR] 22 [18-26] 22.5 [19.3-27] 20 [17-25] 0.015

Cushman score 6 [4-9] 6 [4-9] 7 [4-9] 0.196

SOFA 3 [2-6] 5 [3-8] 3 [1-5] <0.0001

SAPS II 24 [16-34] 27 [17-37] 20 [13-31] 0.0067

4) Table 3: how was persistent confusion at ICU discharge assessed? CAMS-ICU? Clinical impression? How much was delirium vs. dementia? What was baseline confusional status?

Author’s reply: We agree that this point needs to be clarified. For each patient, we used the daily notes of nurses and doctors in the electronic medical record to retrospectively assess the date of AWS resolution or the status of persistent confusion. The date of AWS resolution was the date either where the episode of AWS was stated to have resolved in the EMR, or the last date where AWS was mentioned if it was followed by no further signs or symptoms of AWS for a period of at least 48 hours and no confusion was reported at the time of discharge. If none of these 2 conditions were reported, the episode of AWS was considered as persistent. We agree that this is a non-validated method. However, we provide information on confusion persistence after AWS in the ICU, an area rarely explored.

Six patients (2.9%) had dementia mentioned in their past medical history. For such patients a worsening of the clinical state during hospitalization and at the time of discharge had to be mentioned in the patients’ EMR to classify the patient in the “persistent confusion” category. 

As suggested by the reviewer, we have added the information in the revised version of the manuscript (Methods section and Table 1) as follow:

“For each patient, AWS recovery was assessed by reading the daily notes of nurses and doctors from the EMR. The date of resolution was either the date of resolution recorded in the medical file or the last date of recording of AWS with no further signs or symptoms of AWS recorded for at least 48 hours. If neither of these two conditions was met, the episode of AWS was classified as persistent. When patients had underlying dementia or other neurocognitive disorders, a worsening of the clinical state during hospitalization had to be mentioned in the patients’ EMR to classify a patient with persistent confusion”

Table 1. Baseline characteristics of the 204 study participants

Variable

All patients

 (n = 204)

Complicated hospital stay

(n = 98) 

Uncomplicated hospital stay

 (n = 106) 

P value

Demographics 

Age, median [IQR], years 53 [46-60] 54.5 [48-61] 50 [44-58] 0.099

Male sex, n (%) 172 (84.3) 83 (84.7) 89 (84.0) 0.88

Charlson’s index, median [IQR] 1 [0-3] 2 [0.25-4] 1 [0-3] 0.019

Alcohol withdrawal history 

History of AWS, n (%) 42 (20.6) 23 (23.5) 19 (18.0) 0.387

History of DT, n (%) 10 (4.9) 7 (7.1) 3 (2.8) 0.200

History of withdrawal seizures, n (%) 25 (12.3) 15 (15.3) 10 (9.4) 0.201

Psychiatric history 

Substance use disorder other than alcohol, n (%) 30 (14.7) 12 (12.2) 18 (17.0) 0.429

Any psychiatric disorder, n (%)* 71 (34.8) 33 (33.7) 38 (35.9) 0.598

Mood disorders, n (%) 7 (3.4) 4 (4.1) 3 (2.8) 0.712

Anxiety disorders, n (%) 64 (31.4) 29 (29.6) 35 (33) 0.651

Chronic medications 

Benzodiazepines, n (%) 71 (34.8) 27 (27.6) 44 (41.5) 0.036

Antipsychotic drugs, n (%) 26 (12.8) 6 (6.1) 21 (18.9) 0.0063

ICU admission from the ED, n (%) 137 (67.2) 63 (64.3) 74 (69.8) 0.401

Cushman’s score at ED admission, median [IQR] 7 [4-9] 7 [4-9] 7 [4-9] 0.827

Reason for ICU admission, n (%) 0.00073

Sepsis 61 (29.9) 39 (39.8) 22 (20.8) 

Altered consciousness 60 (29.4) 22 (22.5) 38 (35.9) 

Seizures 24 (11.7) 4 (4.1) 20 (18.9) 

Trauma 20 (9.8) 9 (9.2) 11 (10.4) 

Surgery 12 (5.9) 6 (6.1) 6 (5.7) 

AKI 12 (5.9) 7 (7.2) 5 (4.7) 

Other** 15 (7.4) 11 (11.2) 4 (3.8) 

Clinical variables and measures at ICU admission 

HR, median [IQR], bpm 104 [88-120] 109 [98-123] 99 [85-117] 0.038

SBP, median [IQR], mmHg 125 [106-147] 121 [100-140] 126 [112-150] 0.038

Glasgow Coma Scale score, median [IQR] 14 [12-15] 14 [11-15] 14 [13-15]

 0.84

RR, median [IQR] 22 [18-26] 22.5 [19.3-27] 20 [17-25] 0.015

Cushman score 6 [4-9] 6 [4-9] 7 [4-9] 0.196

SOFA 3 [2-6] 5 [3-8] 3 [1-5] <0.0001

SAPS II 24 [16-34] 27 [17-37] 20 [13-31] 0.0067

AKI: acute kidney injury; AWS: alcohol withdrawal syndrome; BPM: beats per minute; DT: delirium tremens; ED: emergency department; HR: heart rate; ICU: intensive care unit; IQR: interquartile range; RR: respiratory rate; SAPS II: Simplified Acute Physiology Score, version II; SBP: systolic blood pressure; SOFA: Sequential Organ Failure Assessment 

*Any psychiatric disorder, n (%): including 6 patients with underlying dementia

** Other: cardiac or respiratory arrest; upper gastrointestinal hemorrhage; acute pancreatitis; ketoacidosis; mesenteric ischemia

5) Discussion: based on multivariable analysis, authors argue that sicker patients (MOD) have complicated stays. It would be interesting to learn how this compares to patients without AWS.

Author’s reply: We agree that some of our findings in AWS patients (influence of multiple organ dysfunctions on patients’ outcome) may apply to many other diseases or conditions in the ICU setting (Sepsis, ARDS, trauma, pancreatitis,…). As suggested by the reviewer, we collected the available data of patients without AWS admitted to the ICU during the study period. Patients with AWS were younger and had lower severity scores at ICU admission than patients without AWS. The incidence of complicated hospital stay was 48% in patients with AWS and 31% in patients without AWS (p<0.001). Interestingly, the incidence of complicated hospital stay was explained by a higher incidence of extended stay in the ICU while the mortality was lower. In a multivariable model which included age, SAPSII and AWS, AWS had the highest aOR for complicated hospital stay. 

As suggested by the reviewer, this information has been added to the result section of the revised version of the manuscript (page 11 lines 210-211, page 12 line 237, and page 13 lines 238-239). Both tables have been added to the supplementary appendix. 

SA Table 1. Comparison of ICU patients with and without AWS during the study period

Variable

Patients 

with AWS

(n = 204) 

Patients without AWS

 (n =5437) 

P value

Demographics 

Age, median [IQR], years 53 [46-60] 60 [45-70] 0.002

SAPS II, median [IQR], years 24 [16-34] 35 [24-52] 0.001

Outcome 

ICU LOS, median [IQR], days 6 [4-10.3] 3 [2-6] <0.001

ICU LOS≥7days, n (%) 90 (44) 1095 (20.1) <0.001

Hospital mortality, n (%) 16 (7.8) 786 (14.5) 0.008

Complicated hospital stay 

ICU LOS ≥7 days or hospital death, n (%) 98 (48) 1685 (31) <0.001

AWS: alcohol withdrawal syndrome; ICU: intensive care unit; IQR: interquartile range; LOS: length of stay; SAPS II: Simplified Acute Physiology Score, version II

SA Table 2: Logistic regression analyses for factors associated with complicated hospital stay among the 5,641 patients admitted to the ICU during the study period. 

Factors

Multivariable analysis

 OR (95%CI) P value

Age (per year) 0.99 (0.99-1.00) 0.368

SAPS II (per point) 1.06 (1.05-1.06) <0.001

Alcohol withdrawal syndrome 3.53 (2.60-4.81) <0.001

SAPS II: Simplified Acute Physiology Score, version II

Candidate predictors were: Age, SAPS II, and alcohol withdrawal syndrome.

6) Abstract: based on multivariable analysis, authors also argue that seizures are protective. I find this statement illogical since seizures are, by definition, harmful.

Author’s reply: We agree with the reviewer that we need to rephrase the conclusion, which is confusing and could be misinterpreted and misunderstood.

As suggested by the reviewer, we modified the abstract’s conclusion as follow:

“The likelihood of developing complicated hospital stay relied on the reason for ICU admission and the number of organ dysfunctions at ICU admission.”

7) Discussion: the presence of DTs or alcohol withdrawal seizures may suggest opportunities to standardize treatment of AWS in the ED and general wards, which may hopefully reduce incidence of ICU admission.

Author’s reply: We agree with the reviewer that preventing clinical deterioration of AWS by improving its early identification and standardizing its treatment can improve quality of care and patient safety, as reported by Melkonian et al. 

As suggested by the reviewer, this point has been added to the revised version of the manuscript (discussion section – comparison with previous studies – optimal management - page 15 lines 297-300) as follow:

“Interestingly, a recent study reported that the implementation of a hospital-wide protocol for the management of AWS resulted in significant improvements in quality of care, decreased the need for ICU admission and the rate of intubation, reduced hospital length of stay, and was cost-savings (34).”

(34) Melkonian A, Patel R, Magh A, Ferm S, Hwang C. Assessment of a Hospital-Wide CIWA-Ar Protocol for Management of Alcohol Withdrawal Syndrome. Mayo Clin Proc Innov Qual Outcomes. 2019 Aug 23;3(3):344-349. doi: 10.1016/j.mayocpiqo.2019.06.005. eCollection 2019 Sep.

 

Reviewer #2: PONE-D-21-30656 — Alcohol Withdrawal Syndrome in ICU Patients: Clinical Features, Management, and Outcome Predictors

The authors perform a retrospective cohort study of ICU patients with AWS, using manual chart review for data extraction. The main objective was to describe a wide array of factors associated with ICU stay ≥ 7 days and/or in-hospital mortality – a combined outcome the authors labeled “complicated hospital stay.” This is mainly a descriptive study, including descriptions of many patient-level factors stratified by the primary outcome (i.e., complicated hospital stay): demographical characteristics, baseline diagnoses, chronic medications, reason for ICU admission, clinical features associated with acute illness (e.g., SOFA scores), AWS therapies, AWS-related clinical scores (i.e., Cushman) and diagnoses (e.g., seizures), duration of AWS, life-sustaining ICU therapies (e.g., mechanical ventilation), persistent confusion at ICU discharge, length of stay, mortality, and disposition upon hospital discharge. Logistic regression analyses were also performed to further evaluate the association between certain patient factors and the primary composite outcome, but it is not entirely clear how/why the much smaller list of independent variables were selected and included in the model.

The authors should be commended for investigating a grossly understudied yet common ICU condition, offering insights regarding the basic epidemiology of AWS in the ICU and “real world” treatment approaches and hospital course. Although the study is interesting, it lacks focus and does not seem to be driven by a central hypothesis or research question. As a result, the reader gets lost. Many conclusions are stated throughout the discussion that cannot be drawn from this study. Given these broad issues, I am offering general feedback with some specific examples below. Overall, I think the study design and resulting manuscript needs significant restructuring.

Author’s reply: We thank the reviewer for all the comments and for giving us the opportunity to improve our manuscript and to submit a revised version. We intended to conduct an epidemiological study of AWS in the ICU setting, to describe its clinical features, course, treatment and outcome in the ICU. We hypothesized that a proportion of ICU patients with AWS would experience a complicated hospital stay (defined by hospital death or extended stay in the ICU) and we aimed to identify factors associated with such outcomes.

As suggested by the reviewer, the research purpose has been clarified in the revised version of the manuscript and the methods section has been thoroughly revised. 

Examples of issues that need to be addressed:

Lack of transparency re. methods – the data for this study was mainly obtained via manual chart review. This could be a strength if rigorously approached but was incompletely described. As it stands, the methods section leaves many questions unaddressed. For example, were “habits of alcohol consumption” obtained via patient interview? Is the alcohol history part of the standard hospital intake procedure? What was the frequency of missing data (many ICU patients are too sick to provide history)? Did authors AV and EC manually extract ALL data for the study or just confirm the diagnosis of AWS? How were DSM-5 criteria for AWS applied to information recorded in electronic medical records that was not necessarily designed for assessing these criteria? For example, how was “increased hand tremor” (included in the DSM-5 criteria) assessed via the electronic medical records?

Author’s reply: We agree with the reviewer that this point needs to be clarified. Yes, all data were manually extracted from the EMRs for each patient by 2 investigators (AV and EC) to minimize biais and improve accuracy. 

Habits of alcohol consumption and alcohol history are part of the standard intake procedure in our hospital. Data were obtained from patients’ interview. When the patient’s clinical condition made the interview impossible, data were obtained either from the next of kin or from the patient at the time of discharge (when he recovered from the acute illness). However, data on the daily alcohol intake was missing in 51 patients (25%). This information has been added in the revised version of the manuscript (methods and results sections). 

We agree with the reviewer that AWS is a difficult clinical diagnosis with no gold standard criteria. We carefully double checked the plausibility of AWS diagnosis using the combination of ICD10 criteria and DSM-5 criteria. For each patient, a thorough analysis of the electronic medical record (doctors and nurses notes) was done by 2 investigators (AV and EC). All four major criteria had to be present in the electronic medical record of each patient to diagnose AWS. For major criterion B, 2 or more of the 8 symptoms had to be present. Using such methodology, 42 patients were excluded from the analysis because the diagnosis of AWS was unclear (Figure 1, flowchart).

This important point has been clarified in the revised version of the manuscript (methods section) as follow:

“Each medical file was reviewed by AV and EC to confirm the diagnosis of AWS based on the Diagnostic and Statistical Manual of Mental Disorders, Fifth Edition (DSM-5). All four major criteria had to be present in the electronic medical records of each patient to diagnose AWS. For major criterion B, 2 or more of the 8 symptoms had to be present. Data were extracted from the doctors and nurses notes (S1 Fig.)”.

Statistical analysis section is confusing – “Quantitative” and “qualitative” variables are referred to in the first sentence of this section—do the authors mean continuous and categorical variables? It seems a statistical approach (“significant” variables in univariate analyses) versus a hypothesis driven approach was used to design the multivariable logistic regression model. Presumably the authors are using this multivariable model to address confounding, but the underlying hypothesis regarding how these variables relate is unclear. 

Author’s reply: We agree with the reviewer that our statistical section is unclear and needs to be clarified. As suggested by the reviewer, “quantitative” and “qualitative” have been replaced by “continuous” and “categorical” variables in the revised version of the manuscript. 

Most of the literature on AWS focused on identifying predictors of delirium tremens (DT) or seizures. However, whether DT or seizures are associated with other relevant patients-centered outcomes (mortality, extended ICU stay) remains unclear. Therefore, we purposefully chose to identify factors associated with hospital death or extended ICU stay (combined outcome analyzed as a binary variable), an area almost unstudied in ICU patients with AWS. Our aim was to identify frontline variables (available at the time of ICU admission) associated with such outcomes to help clinicians for early identification of patients at risk of clinical deterioration who may benefit the most from close monitoring and therapeutic interventions. 

Therefore, we purposefully preselected variables which plausibly fit these outcomes based on knowledge from the literature (SOFA, comorbidities, and mortality) and assumptions (chronic use of BZD or antipsychotics, history of AWS, reason for ICU admission, and extended stay in the ICU). We carefully checked to avoid collinearity (for example: SOFA and heart rate, blood pressure, and respiratory rate) and we applied the rule of selecting a maximum of 1 variable per 8 events (total of 12 variables in our study). All variables included in the model are displayed and thus, we believe our assumptions are transparent and explicit. 

As suggested by the reviewer, we have revised the statistical methods section as follow:

“Continuous variables are described as median and interquartile range [IQR] and compared using Wilcoxon’s test. Categorical variables are described as counts (percent) and compared using the exact Fisher’s test. The occurrence of complicated hospital stay (versus uncomplicated hospital stay) was analyzed as a binary variable. Logistic regression analyses were performed to identify variables associated with complicated hospital stay, with estimated odds ratios (ORs) and their 95% confidence intervals (95%CIs). For the multivariable model, we preselected candidate variables which plausibly fit with complicated hospital stay based on knowledge from the literature (SOFA, comorbidities, and mortality) and our assumptions (chronic use of BZD or antipsychotics, history of AWS, reason for ICU admission, and extended stay in the ICU). We carefully checked to avoid collinearity between variables and we applied the rule of selecting a maximum of 1 variable per 8 events (total of 12 variables in our study). All tests were two-sided, and P values lower than 5% were considered to indicate significant associations. Statistical tests were conducted using the R statistics program, version 3.5.0 (R Foundation for Statistical Computing, Vienna, Austria; www.R-project.org/).” 

In addition, preselected candidate variables included in the multivariable model are now clearly stated in the footnote of the revised version of table 4.

An excellent reference for thinking about study design and presentation of results is: Lederer et al. Annals Am Thorac Soc 2019;16(1):22-28.

“Overreaching” conclusions – In the final paragraph (and similarly stated in the abstract), “AWS in ICU patients chiefly affects young patients with few comorbidities and is often triggered by a precipitating factor such as sepsis, trauma, or surgery.” A more accurate statement from my perspective might be: “ICU patients in this sample drawn from a single hospital in France were predominantly male (84%) with a median age of 53 (IQR 46-60) and were commonly admitted with additional comorbidities including sepsis, trauma, or following (elective?) surgery.” We do not know that AWS was “triggered” by comorbid conditions like sepsis. AWS is more likely triggered by heavy alcohol use that is ALSO possibly associated with these other conditions (based on data from other studies).

Author’s reply: We agree that our conclusions may go beyond our results and should be rephrased and tempered. We fully agree that no causal effect can be drawn from our study. However we suggest swapping “additional comorbidities” by “additional diagnoses” to avoid confusion between comorbidities (diabetes, hypertension,…) and acute conditions (sepsis, surgery,…). 

As suggested by the reviewer, we have modified the abstract conclusion and the final paragraph of the manuscript as follow:

Abstract conclusion 

“AWS in ICU patients chiefly affects young adults and is often associated with additional factors such as sepsis, trauma, or surgery. Half the patients experienced an extended ICU stay or death during the hospital stay. The likelihood of developing complicated hospital stay relied on the reason for ICU admission and the number of organ dysfunctions at ICU admission.”

Manuscript conclusion 

“ICU patients in this sample drawn from a single hospital in France were predominantly male (84%) with a median age of 53 (IQR 46-60) and were commonly admitted with additional diagnoses including sepsis, trauma, or following elective or urgent surgery.”

Implicit comparisons to a larger ICU sample, not included in the study – The authors seem to make comparisons to a broader ICU cohort. For example, “Despite having low severity scores at ICU admission”… or “the high frequency of persistent confusion” – these statements imply comparisons but the comparison group (implicitly, ICU patients at large) is not defined for the reader. The authors’ tendency to make such comparisons illustrates perhaps the fundamental design flaw of the study. Descriptions of ICU patients with AWS are provided, but without context. The reader is left wondering, how does this compare to “average” ICU patients at the study hospital? The association identified between organ dysfunction and the combined outcome of ICU stay ≥ 7 days and/or in-hospital mortality is not surprising in ICU patients. Whether or not AWS modifies this relationship would be the interesting question; for example, testing the hypothesis that the known association between organ dysfunction and ICU length of stay and/or in-hospital mortality is more pronounced in patients with AWS compared to patients without AWS.

Author’s reply: We agree that some of our findings in AWS patients (influence of multiple organ dysfunctions on patients’ outcome) may apply to many other diseases or conditions in the ICU setting (Sepsis, ARDS, trauma, pancreatitis,…). As suggested by the reviewer, we collected the available data of patients without AWS admitted to the ICU during the study period. Patients with AWS were younger and had lower severity scores at ICU admission than patients without AWS. The incidence of complicated hospital stay was 48% in patients with AWS and 31% in patients without AWS (p<0.001). Interestingly, the incidence of complicated hospital stay was explained by a higher incidence of extended stay in the ICU while the mortality was lower. In a multivariable model which included age, SAPSII and AWS, AWS had the highest aOR for complicated hospital stay. 

As suggested by the reviewer, this information has been added to the result section of the revised version of the manuscript (page 11 lines 210-211, page 12 line 237, and page 13 lines 238-239). Both tables have been added to the supplementary appendix. 

SA Table 1. Comparison of ICU patients with and without AWS during the study period

Variable

Patients 

with AWS

(n = 204) 

Patients without AWS

 (n =5437) 

P value

Demographics 

Age, median [IQR], years 53 [46-60] 60 [45-70] 0.002

SAPS II, median [IQR], years 24 [16-34] 35 [24-52] 0.001

Outcome 

ICU LOS, median [IQR], days 6 [4-10.3] 3 [2-6] <0.001

ICU LOS≥7days, n (%) 90 (44) 1095 (20.1) <0.001

Hospital mortality, n (%) 16 (7.8) 786 (14.5) 0.008

Complicated hospital stay 

ICU LOS ≥7 days or hospital death, n (%) 98 (48) 1685 (31) <0.001

AWS: alcohol withdrawal syndrome; ICU: intensive care unit; IQR: interquartile range; LOS: length of stay; SAPS II: Simplified Acute Physiology Score, version II

SA Table 2: Logistic regression analyses for factors associated with complicated hospital stay among the 5,641 patients admitted to the ICU during the study period. 

Factors

Multivariable analysis

 OR (95%CI) P value

Age (per year) 0.99 (0.99-1.00) 0.368

SAPS II (per point) 1.06 (1.05-1.06) <0.001

Alcohol withdrawal syndrome 3.53 (2.60-4.81) <0.001

SAPS II: Simplified Acute Physiology Score, version II

Candidate predictors were: Age, SAPS II, and alcohol withdrawal syndrome.

---

## [Editor Report · Decision Letter 1]

2 Dec 2021

Alcohol Withdrawal Syndrome in ICU Patients: Clinical Features, Management, and Outcome Predictors

PONE-D-21-30656R1

Dear Dr. Canet,

We’re pleased to inform you that your manuscript has been judged scientifically suitable for publication and will be formally accepted for publication once it meets all outstanding technical requirements.

Kind regards,

Aleksandar R. Zivkovic

Academic Editor

PLOS ONE

---

## [Editor Report · Acceptance letter]

9 Dec 2021

PONE-D-21-30656R1 

Alcohol withdrawal syndrome in ICU patients: clinical features, management, and outcome predictors 

Dear Dr. Canet:

I'm pleased to inform you that your manuscript has been deemed suitable for publication in PLOS ONE. Congratulations! Your manuscript is now with our production department. 

Kind regards, 

on behalf of

Dr. Aleksandar R. Zivkovic 

Academic Editor

PLOS ONE